# Effect of Water Supplementation on Oxidant/Antioxidant Activities and Total Phenol Content in Growing Olives of the Morisca and Manzanilla Varieties

**DOI:** 10.3390/antiox11040729

**Published:** 2022-04-07

**Authors:** Inmaculada Garrido, Marcos Hernández, José Luis Llerena, Francisco Espinosa

**Affiliations:** 1Research Group FBCMP(BBB015), Faculty of Sciences, Campus Avenida de Elvas, University of Extremadura, s/n, 06006 Badajoz, Spain; igarridoc@unex.es (I.G.); jllerena@unex.es (J.L.L.); 2CIFP La Granja, Government of Cantabria, 39792 Heras, Spain; mhernandezs06@educantabria.es; 3Agri-Food Technological Center of Extremadura (CTAEX), Ctra. Villafranco-Balboa 1.2, 06195 Villafranco del Guadiana, Spain

**Keywords:** irrigation, *Olea europaea*, peroxidase, polyphenol oxidase, ripening, superoxide dismutase

## Abstract

The objective of this work was to analyse, using a time series analysis, the effect of water regime for two cultivars at three stages of ripeness, during three consecutive years. Fruit and oil yield; O_2_^−^ production and NADH oxidation activities; polyphenol oxidase (PPO), superoxide dismutase (SOD) and peroxidase (POX) activities; total phenols, flavonoid and phenylpropanoid glycoside content; and total antioxidant capacity (FRAP) were determined. All these parameters were found to depend on variety, irrigation and year. The results showed that the fruit and oil yields were strongly dependent on both irrigation and variety. The DW/FW ratio was practically constant during ripening, with small variety-dependent changes due to irrigation. Total amino acid and protein contents increased with ripening, with a close dependence on variety but not on irrigation. The SOD and POX activities appeared closely related, and related to the NADH oxidation and the amount of O_2_^−^. The evolution of phenols and FRAP during ripening was complementary to that of NADH oxidation, O_2_^−^ production as well as SOD and POX activities. The determining factors of the SOD, POX and PPO activities were the variety and the ripening; the determining factor of the yield, ROS production, total phenols and antioxidant capacity was the water regime. Inverse correlations were observed between maximum temperature and total phenols (−0.869), total flavonoids (−0.823), total PPGs (−0.801) and FRAP (−0.829); and between DW/FW and irrigation (−0.483). The remaining significant correlations were positive.

## 1. Introduction

Virgin olive oil (VOO) contains a large amount of antioxidant compounds which relieve the oxidative stress caused by free radicals, thereby having negative health effects such as cancer, cell damage, atherosclerosis [1,2,3]. Most of these beneficial effects of VOO come from phenolic compounds [4,5,6]. The quality of the oils is closely related to the cultivar, the production region and the farming practices carried out during their production. From this perspective, monovarietal VOOs have been shown to be differentiated on the basis their compositional characteristics, and there are also differences originating in the environmental and growing conditions [7,8,9]. This is especially true for the varieties and locations that have PDO (Protected Designation of Origin) or PGI (Protected Geographical Indication) certification.

Traditionally, olive groves in the Mediterranean basin were purely rainfed, and their production depended on the weather conditions which decisively influenced the organoleptic properties of their oils [3]. This fact is of especial interest for olive varieties whose behaviour is adapted to their production environment, as is the case of the Morisca and Manzanilla varieties of great interest in Extremadura, Spain [8]. These and other varieties make it possible to obtain monovarietal oils with characteristics that clearly differentiate them from other more widely established varieties.

Currently, olive production is expanding into areas in which this crop has not traditionally been grown. Areas of high production can be expected to change even more, with such changes inducing a profile of different metabolites that will give their VOOs an individual character [10]. This aspect is especially important with the introduction of irrigation, which can modify the productivity, water content, phenolic composition and antioxidant capacity, depending largely on the variety [3]. Only a few varieties, such as Arbequina, Arbosana and Koroneiki, can be used for intensive (more than 1500 olive trees/ha) olive production [11,12,13]. The expansion of the use of these techniques has naturally led to a standardization of production, with the consequent loss of the particular qualities of VOO that can be provided by all the other varieties that are not apt for intensive cultivation. Clear differences in the antioxidant compound content of VOOs have been found to be due to the combined effect of temperature, insolation and aeration, and to the availability of water in different soils [3,14].

The ripening of olives is a very complex process which involves the development of multiple changes and metabolic processes in the fruit, leading to softening, change in texture and increased oil content. Olives have a high content of phenolics. These evolve with ripening, and have strong antioxidant activity [15,16,17]. The stage of ripening of the fruit is considered to be a highly influential factor determining the phenolic composition of the oil, so it is interesting to monitor the effect that ripening has on the final product. During ripening, biochemical processes occur in the fruit that affect its content of sugars, phenols etc., and these processes lead to changes in texture, firmness and colour which in turn determine the fruit’s nutritional and organoleptic quality [18,19]. Changes during ripening occur at the membrane level. They end in a process of programmed cell death. Fruit ripening involves an oxidative process, and modulation of the antioxidant system [19,20]. All this makes ripening a complex system of interactions that defines the final characteristics of the oil that is obtained.

The objective of the present study was therefore to analyse the effect of the water regime (with or without irrigation) in two olive cultivars (Manzanilla and Morisca), at three ripening stages (S1, S2, S3) and during three consecutive years. This influence was studied through the changes in the parameters of fruit and oil production; in the phenol, flavonoid and phenylpropanoid glycoside content; in the O_2_^−^ production; and in SOD, POX and PPO activities, all of which are related to the physiological process of ripening and the response to stress. This study is complemented with correlation and time-series analyses of the influence of the climate and the production regime variables on the investigated parameters. Within linear processes, a simple additive model was applied as method of time-series analysis.

## 2. Materials and Methods

### 2.1. Olive Sampling and Sample Preparation

The experiment was carried out in Ribera del Fresno (Badajoz, Spain) in two olive groves (*Olea europaea*, L.; cv. Manzanilla de Sevilla and cv. Morisca). The trees were 30 years old, and both groves are maintained under conditions of no-tillage, post-emergence herbicide weed control and traditional pruning. The soil type of both plots is sandy clay loam. The climate is Mediterranean, and the local conditions are summarized in Appendix A [21].

Two types of water regime were studied—irrigation (FI) in accordance with the levels of crop evapotranspiration, and rainfed (NI). In the irrigated grove (FI), water was applied in accordance with the trees’ theoretical requirements calculated using ETo and the crop coefficient Kc. The irrigation period was from 1 April to 31 October (Appendix A).

Olives at the same stage of colour change were harvested at random from around the canopy of ten trees of each variety (Manzanilla/Morisca) and each experimental condition (NI/FI), at a height of 1.5 m. The ripening stages sampled were green (S1, 0–1), veraison (S2, 2–3) and black or mature (S3, >3) [22], with the sampling done in September (week 2), November (week 2) and December (week 1), respectively, from 2011 to 2013. The fruits were picked before 10 a.m. local time, and immediately frozen in liquid nitrogen until biochemical analysis. The oil concentration (% dry weight) was determined by near-infrared (NIR) spectroscopy (Olivesan equipe FOSS), and the fruit yield (kg ha^−1^) and total oil production per hectare were calculated.

### 2.2. Determination of the Dry Weight/Fresh Weight Ratio, and the Soluble Amino Acid and Protein Contents

The olive pulp was first weighed fresh, then oven-dried at 90 °C for 24 h followed by measurement of the dry weight and calculation of the dry weight/fresh weight (DW/FW) ratio. A 0.2 g aliquot of the pulp was homogenized in 1 mL of 50 mM sodium phosphate buffer (pH 7.0), then filtered through muslin, centrifuged at 12,360× *g* for 15 min at 4 °C and the supernatant used for the protein and amino acid assays. The protein content was determined by the Bradford method [23]. The amino acid content was determined in accordance with Yemm and Cooking [24]. Briefly, 100 µL of supernatant was mixed with 1.5 mL of ninhydrin reagent, and incubated at 100 °C for 20 min. The mixture was cooled, and 8 mL of 50% propanol was added. The result was left at room temperature for 30 min, and then the A_570_ was measured. The results are expressed as mg of amino acids g^−1^ FW against a glycine standard curve.

### 2.3. Total Phenolics, Flavonoids and Phenylpropanoid Glycosides (PPGs)

The olive pulp aliquots were homogenized in methanol, chloroform and 1% NaCl (1:1:0.5). The homogenate was filtered and centrifuged at 3200× *g* for 10 min at 4 °C, and the methanol phase separated for the phenolic compound assay. Total phenolics (expressed as µg caffeic acid g^−1^ FW) were determined spectrophotometrically (Shimadzu 1603, Kyoto, Japan) at A_765_ with Folin–Ciocalteu reagent [25]. Total flavonoids (expressed as µg rutin g^−1^ FW) were determined at A_415_ in accordance with Kim et al. [26], calculating the content on the basis of the rutin standard curve. The PPG levels (expressed as µg verbascoside g^−1^ FW) were determined at A_525_ based on estimating an *o*-dihydroxycinnamic derivative using Arnow reagent [27].

### 2.4. Oxidant/Antioxidant Enzyme Activities

Enzyme activities were determined on an extract of the raw olives. For the PPO activity, the pulp of the olive samples was homogenized at 4 °C at a concentration of 0.15 g mL^−1^ in 100 mM phosphate buffer (pH 7.0), 1% PVPP. The homogenate was filtered and centrifuged at 12,000× *g* for 15 min at 4 °C. The filtered supernatant was immediately used for assay. For the other enzymes, the pulp of the raw olives was homogenized at 4 °C at a concentration of 0.5 g mL^−1^ in 50 mM phosphate buffer (pH 6.0). The homogenate was filtered and centrifuged at 39,000× *g* for 30 min at 4 °C. The pellet was discarded, and the supernatant filtered for the assays and protein content determination [23].

NADH oxidation was measured in 50 mM phosphate buffer (pH 6.0), 300 µM NADH, and the enzyme extract by the fall in A_340_ [28] (ε = 6.3 mM^−1^ cm^−1^), expressed as nmol NADH_ox_ min^−1^ mg^−1^ prot. The O_2_^−^ generating activity was measured spectrophotometrically for the oxidation of epinephrine to adrenochrome at A_480_ (ε = 4.020 mM^−1^ cm^−1^) [29,30]. The reaction mixture contained 1 mM epinephrine in 25 mM acetate buffer (pH 5.0) and the enzyme extract. The result is expressed as nmol adrenochrome min^−1^ mg^−1^ prot. SOD (EC 1.15.1.1) activity was determined at A_560_ in 50 mM phosphate buffer (pH 7.8), 0.1 mM EDTA, 1.3 µM riboflavin, 13 mM methionine, 63 µM NBT and the enzyme extract [31]. A unit of SOD is defined as the amount of enzyme required to cause 50% inhibition of NBT reduction. POX (EC 1.11.1.7) activity was measured at A_590_ (ε = 47.6 mM^−1^ cm^−1^) in a reaction medium with 3.3 mM DMAB and 66.6 µM MBTH in 50 mM phosphate buffer (pH 6.0) and the enzyme extract [32]. The result is expressed as nmol DMAB-MBTH (indamine dye) min^−1^ mg^−1^ prot. PPO (EC 1.14.18.1) activity was determined at A_390_ in 100 mM phosphate buffer (pH 6.5), 0.58% (*v*/*v*) Triton X-100, 30 µM caffeic acid and the enzyme extract [33]. A unit of PPO is defined as the amount of enzyme required to cause a decrease in absorption of 0.001 units min^−1^.

### 2.5. Total Antioxidant Activity Assay (FRAP)

The ferric reducing ability of plasma (FRAP) was performed in accordance with Rios et al. [34]. Olive samples were homogenized with methanol (0.10 g mL^−1^). The homogenate was filtered and centrifuged at 10,000× *g* for 2 min at 4 °C. Then, 10 µL of the homogenate was mixed with 1500 µL FRAP reagent, left at room temperature for 5 min, and then measured at A_593_. Calibration was done against a standard curve using freshly prepared ferrous ammonium sulphate, and the result expressed as μg of ferrous sulphate g^−1^ FW.

### 2.6. Data Analysis

Spider graphs were applied to compare each variety’s productive parameter profiles according to the water management regime. The correlations between the chemical compound and climatological data were calculated. We conducted a slightly modified additive-type time-series analysis, using the arithmetic mean instead of moving averages. A time series is a set of observations, each of which is recorded at a specific time. For correct interpretation, it is important to remove the presence of seasonal components in a process known as seasonal adjustment. The classical decomposition model was applied [35], in which each datum X_t_ is represented as combination of processes, X_t_ = m_t_ + s_t_ + Y_t_, where m_t_ is a slowly changing function (the trend component), s_t_ is a function with known period (the seasonal component) and Y_t_ is a random noise component. In other words, the measured numerical value of a variable is the sum of its intrinsic value and the effect of the environment or circumstances in which it occurs. This concept is shared by other statistical techniques, an example being the concept of communality in principal component analysis. The result of this analysis is a set of graphs that define the de-seasoned behaviour profile of each of these intrinsic variables, allowing them to be compared for the three agronomic variables—variety, ripening stage and water management regime. The time-series analysis was performed using Microsoft Excel 16.49, and the correlation analysis between the seasonal variation indices and the environmental variables using SPSS 21.0 (SPSS Inc., Chicago, IL, USA).

## 3. Results and Discussion

The impact of the water regime and the response of each variety through the three seasons studied can be analysed using the spider graphs constructed from the raw data (Appendix A) [21]. Figure 1 shows the values for ripening stage S3, collected in the first week of December. They were clearly affected by the environmental conditions of the two preceding months. As can be seen in Appendix A, being autumn months with high water inputs, they are the months (after the formation of the fruit) with the lowest ETo values, especially in comparison with the summer months of July, August and September. This is the reason for the major influence of the water management regime, and the interest in knowing its effect on the physiological and production parameters.

With regard to the variable representing the concentration of oil (expressed as % of DW), the Morisca variety showed little year-to-year variation for both types of production (NI and FI), with the greatest value corresponding to 2013 and the lowest to 2011. For the Manzanilla variety, this parameter was relatively constant both between production regimes and from year to year.

The fruit and the oil production parameters are closely linked. For the Morisca variety, the greatest productions of olives corresponded to 2011 (with the values being much greater for FI than for NI), while 2013 had the lowest production of both olives and oil. For the Manzanilla variety, there was increased oil production with irrigation in 2011 and 2013. This suggests that water management not only affects the size of the olive, but also has a direct effect on its capacity to accumulate oil. These results are in agreement with those obtained by Fernandez-Silva et al. [36] on the Cobrançosa variety in which irrigation input increases olive and oil yield, but does not alter oil content. However, for the Frantoio variety, Cirilli et al. [37] describe how irrigation input increases fruit and oil yield, as well as oil content. In any case, the studies were carried out with different varieties—a factor that largely determines these yields.

Table 1 presents the average composition of the olive samples and the factors that significantly influenced those compositions. Ripeness was the factor with the most significant influence on the various parameters, with the protein content having especial relevance (the lowest *p*-value, i.e., the most statistically significant). The individual values corresponding to each variety and water regime are shown in Figure 2, Figure 3 and Figure 4.

Table 2 lists the overall mean values for each parameter analysed for the three years of sampling. The table also lists the seasonal variation indices—that is, the differences with respect to the values expected for that year. Negative values indicate that obtained values were lower than expected, and positive values indicate that values were higher than expected; the closer the value is to zero, the smaller the variation in the quantity. Thus, in 2011, the compounds with antioxidant capacity (e.g., phenols, flavonoids and PPGs) and the FRAP capacity were below the expected values, as were the amount of O_2_^−^ produced, the NADH oxidation and the SOD and POD activities. In 2012, the PPO activity presented a lower-than-expected value. Finally, in 2013, the phenolics, the FRAP and the SOD activity were above the expected values, while the oxidative activities, the protein and amino acid contents and the DW/FW ratio were below.

Different cultivars can adapt differently to changes in climate and cultivation practices [38,39]. The multiple interactions between phenolic compounds, antioxidant activities, antioxidant capacity, rainfall, temperatures, insolation, soils etc. are presented with correlation coefficients in Table 3 (derived from Appendix A) [21].

In these results, particularly notable are the significant inverse correlations of phenols, flavonoids, PPGs and FRAP with the minimum temperatures, and of supplementary water (FI) with the DW/FW ratio; these results confirm that higher olive water content is associated with a lower DW/FW ratio. Fruit and oil yields increased, with oil content (% DW) remaining very similar [3,36].

With respect to the correlations between oxidant and antioxidant compounds, the production of reactive oxygen species (ROS) is a key part of both plants’ response to stress and of physiological processes such as fruit ripening [40,41,42,43]. The cell’s redox state depends on ROS levels [19,44,45]. Enzymatic systems such as those of SOD and POX are involved in this redox control, as are non-enzymatic systems such as those of the phenolics [46,47,48]. These non-enzymatic systems are closely related to PPO activity, which takes part in the hydroxylation of monophenols to diphenols, and in the oxidation of diphenols to quinones [49]. In this process, H_2_O_2_ is formed which, together with phenols, can constitute substrates for POX [50]. The interaction between POX and PPO activities may determine the phenol content of olive oils, not just during ripening but also, due to the release of phenols, during milling [51]. The SOD and POX activities are closely related to the stage of ripeness [41,52].

Each compound can be analysed individually using time-series plots, comparing the shape of the midline and the distribution of individuals on each side of the midline. The DW/FW ratio (Figure 2) remained very stable throughout ripening, with a final increase at S3. The determining factor in this ratio was water supplementation. In all stages of ripening, the ratio was greater in NI because these olives had a lower water content. The values are grouped according to the water management regime (NI/FI), independently of the variety. With respect to protein content and total soluble amino acids, similar behaviours were observed for the two cases, with increases depending on ripening and variety. These compounds group in accordance with the two varieties, independently of the water management regime. This shows that these parameters depended on the variety studied and that they evolved with the ripening of the fruit. The amino acid content of the Morisca variety was greater than that of Manzanilla. This difference between varieties was less marked in the ripest state (S3). The DW/FW ratio remained practically constant throughout all ripening stages, without any variety-dependent changes, and only a slight influence of the water regime factor. With regard to the total content of soluble amino acids and proteins (Figure 2), the behaviour observed was similar, with an increase as ripening advances, reaching the highest levels at S3. These results contrast with those for the Arbequina and Picual varieties reported by Zamora et al. [53] who, for a single cropping year, observed no increases in protein content during ripening. Nonetheless, the data described by those workers would be very similar to ours if we were to only consider the behaviour of these parameters either in the 2013 campaign in isolation for the two varieties, or for the Morisca variety in 2012. Profound changes in the protein content of a given variety’s olives have been described depending on the year factor, or even in response to different geographical locations and therefore different environmental conditions [54]. Ortega-García et al. [55] for cv. Picual and Ebrahimzadeh et al. [54] for cv. Zard describe a clear increase in protein content with ripening, in agreement with our results. Regardless of the state of ripening, cv. Morisca had a greater content of both soluble amino acids and proteins than cv. Manzanilla, without the water management factor having any significant impact on these differences.

In the three years of study, under rainfed conditions, NADH oxidation increased in the S2 state with respect to S1 for both varieties and then decreased at S3. Under irrigation conditions, from S1 to S2 the behaviour was similar, but there was no decrease from S2 to S3. This seems to indicate that in our case this process did not depend on the variety, but did depend on the stage of ripening and the added water supply. At S1 and S2, NADH oxidation clearly depended on the water management factor, with greater values in NI than in FI, with the opposite being the case at S3 for which greater values occurred under FI conditions. These results indicate a dependence of this process on both ripeness stage and irrigation, but not on the variety. As was to be expected, the results of O_2_^−^ production were similar (Figure 3). The irrigation factor was key, determining greater O_2_^−^ production at S1 and S2 (at which stage the maximum was reached), and declining at S3, being greater in FI than in NI. The results show how the rainfed regime induced an increase in O_2_^−^ production as a consequence of the water deficit, which was not observed in FI. At S3, the influence of the ripening factor was greater; indeed, it was the determining factor in O_2_^−^ content at this stage. Comparing the two varieties, the Manzanilla variety presented lower levels of O_2_^−^ production than Morisca. Rbohs (respiratory burst oxidase homologues, plasmalemma NADH oxidases) are responsible for the production of O_2_^−^, consuming NADH in the process. There is increased production of ROS (including O_2_^−^) in the fruits at the intermediate stages of ripening, followed by a final decline [56,57]. These compounds are related to both the physiological process of ripening and the response to environmental stressors (ROS) [42,57,58,59,60]. The difference in response depending on the water management conditions may be evidence for the development of stressors influencing these oxidative processes. Increases in the production of these radicals are related to the ripening process and to the stress response.

The SOD activity showed a strong dependence on the variety, with greater activity in Manzanilla than in Morisca (Figure 3), but no dependence on the water management regime (rainfed or irrigated), since the results were similar under the two conditions for both varieties. With respect to the influence of the stage of ripening, while in Manzanilla the values remained very stable with no appreciable changes, in Morisca the activity increased with ripening to very similar values at S2 and S3. In the rainfed regime, SOD increased with ripening from S1 to S2, followed by a slight decrease at the fully ripe state, S3. In the irrigated regime, this final decline was either less pronounced or absent. The behaviour and factor dependence of the POX activity (Figure 3) were very similar to those of SOD. For cv. Manzanilla, there were no differences in response to growing conditions, although for the rainfed conditions this activity was greater at S2 than at S1 followed by a decline at S3. On the contrary, Morisca showed differences in POX activity between the rainfed and irrigated regimes, although there was a greater dependence on the stage of ripening, with greater activity at S2 for both water management regimes. The contribution of irrigation was limited, with just small increases in activity due to rainfed management in cv. Manzanilla and no increases in cv. Morisca. These results are very similar to those that have been reported for the Gordal and Manzanilla varieties [61], with greater SOD activity in Gordal, which is more susceptible to oxidative stress. As another example, a similar increase in SOD activity has been described in the Picual variety from S1 to S2, with a decrease at S3 back to values very similar to those at S1 [20]. The increase in SOD activity would be indicative of a response of the antioxidant defence system to alterations in the oxidative state that occur during ripening [20]. In our study, the Morisca variety presented the greater amount of O_2_^−^ and the lower SOD activity. On the contrary, in cv. Manzanilla (which has greater SOD activity) the O_2_^−^ production levels were lower, possibly due to more efficient elimination by SOD. It may be that in Morisca there intervenes some other system of defence against oxidative damage (e.g., phenolics), or that this variety is less sensitive.

As was noted above, there were no irrigation-dependent alterations in POX activity, but there were changes dependent on the variety and the stage of ripening. The Manzanilla variety presented greater POX activity at S1 and S3, with the values being similar to cv. Morisca at stage S3. In the varieties Arbequina [62], Picual and Arbequina [51], the behaviours reported are similar to those described here. On the contrary, in the Frantoio variety [37], clear decreases in POX activity have been observed during ripening, ending with a final rise; this was observed together with a strong dependence on irrigation, with the POX activity being lower under deficit irrigation conditions. These changes may be related to the increases that occur in the amount of O_2_^−^ through ripening, with greater or lesser SOD and POX activities occurring depending on those increases.

The SOD and POD antioxidant activities are involved in the processes that control redox balance [63]. In particular, they control the levels of ROS during the different phases of ripening and in response to the environmental stresses with which the fruits are faced during their development [20]. In our case, SOD and POX appeared to be closely related, with fluctuations similar and related to the oxidation of NADH and the amount of O_2_^−^. The phenolic profile of the olives changes with ripening, and in this process POX intervenes by oxidizing phenolic glycosides [55]. The cv. Manzanilla variety showed higher SOD and POX activity than Morisca. The influence of the genetic factor is key, and determines the behaviour of these activities [61]. The influence of environmental factors such as the cultivation regime was reflected in higher SOD and POX activity levels under rainfed conditions. While the cultivation regime affected the evolution of these activities during ripening, the factors that most determined the behaviour of SOD and POX were ripening and variety.

In summary, NADH oxidation and O_2_^−^ production grouped in accordance with the water management regime. The behaviour of SOD was flatter, with the lowest level at S1, and the levels at S2 and S3 being very similar. Despite this gradual increase in SOD activity with ripening, the variety factor was the most influential, with there being very different activity levels between the two varieties regardless of other factors such as irrigation (which naturally also had an influence). In all cases, we observed much greater values in the Manzanilla variety than in the Morisca variety, again independently of the irrigation regime. POX activity was dependent on the variety or the stage of ripening, except in S3 where the water regime made the difference.

In the three years as a whole, phenols, flavonoids and total PPGs (Figure 4) decreased from S1 to S2 and increased at S3. There was a strong dependence of the phenol content on water supplementation; it was greater under NI conditions. A similar result of water stress on phenolic content was described by Petridis et al. [64] in Greek olive varieties. The flavonoid content was lowest at S2 for both varieties and for both growing conditions. The greatest total flavonoid content was observed at S3 under NI conditions, being especially marked for cv. Morisca. Additionally, the PPG content was greater in both varieties for the rainfed conditions. The PPG content was influenced by variety at S1, with a greater content corresponding to cv. Morisca, while at S2 and S3 the values depended completely on the water management regime, and not on the variety.

There was a varietal dependence of PPO activity; it was greater in Morisca (Figure 4). Irrigation conditions also significantly affected this activity at S1 and S2, with the observed grouping being first by variety and then, within each variety, by water management regime, with the greater value corresponding to the rainfed conditions.

The determining factor of phenolic content has been a subject of debate. Špika et al. [10] observed that the phenolic content depended mainly on the variety, but also on rainfall. On the contrary, Uylaşer [17] and Ripa et al. [65] indicated that the environment was the factor determining phenolic composition, with little influence of the genotype.

In this study, phenolic contents increased with ripening; the greatest values were found at S3. Total phenolics depended only on the ripening stage. Total flavonoid and PPG contents were influenced by variety, supply of water and ripening stage. The Morisca variety had greater amounts of flavonoids and PPGs than Manzanilla. In both varieties, the rainfed regime induced greater contents of these compounds. Cerretani et al. [66] described a decrease in phenolic content with ripening, but in the present study both Morisca and Manzanilla showed a decline from S1 to S2 but a rise to S3. These results coincide with the increase in phenol content with ripening described for the Picual variety by López-Huerta and Palma [20], although in that case the increase occurred continuously without the decrease at S2 observed in our varieties. Again, the variety was the key factor in the behaviour of these compounds. The Picual variety cultivated under rainfed conditions has been shown to present oscillations in phenolic content as the olives ripen [55]. On the contrary, in Arbequina, Frantoio, Picual and Verdial [67], and Arbequina, Farga and Morrut [68], decreases in phenolic content have been observed with ripening. Fluctuations in flavonoid content in response to ripening have also been reported [68], with decreases in the final content and large differences between varieties. Slight decreases in the phenol and flavonoid contents with ripening have been described in the Manzanilla and Kalamata varieties [69]. The evolution of the phenolic compound content has been shown to depend on the stage of ripening, the variety and the environmental and cultivation conditions.

Here, the Morisca variety had greater PPO activity than the Manzanilla variety. The evolution of PPO with ripening also depended on the variety. In Morisca, PPO decreased from the beginning of ripening to its completion at S3; this result is similar to what has been observed for the Gordal and Picual varieties [61]. On the contrary, in Manzanilla, the PPO remained fairly stable throughout ripening, with just a slight final rise. This varietal dependence of PPO has also been reported for the Picual and Arbequina varieties [51]. Fluctuations in PPO activity with ripening have been described in the Chétoui variety, as have decreases in Arbequina [40,61] and increases in the varieties Zard [54] and Frantoio [37]. This activity was found to be influenced by both the variety and the ripening stage, while the irrigation conditions had little influence.

The SOD, POX and PPO activities are the bases of the maintenance of redox homeostasis and the phenolic profile during ripening. Thus, POX and PPO together modulate the fruits’ ripening and their final phenolic characteristics—characteristics which are crucial for the quality of the oils that will ultimately be obtained. The POX and PPO activities were closely related to each other, with a dependence on ripening stage. It may be that PPO activity is more strongly related to ripening since it is unaffected by the cultivation conditions, whereas POX seems to be affected by both.

At S3, the two varieties coincided in their response to the water regime, indicating that at this stage of ripening the irrigation factor is more influential than the variety factor, since both behaved in a similar way with only a dependence on the irrigation factor. The case at S1 was similar, although at this stage only the NI conditions coincided for the two varieties while the extra irrigation contribution in FI modified their behaviour. It is especially important to underscore the complete agreement shown by the total phenol, flavonoid and phenylpropanoid glycoside contents, all of which showed the same trend. Similar behaviour was observed for PPO, for which the highest levels of activity were at S1 and S2 where the most influential factor was the variety. Within varieties, the irrigation factor was the most influential. On the contrary, at S3 the most influential factor was again NI/FI, regardless of the variety.

The ripening stage and the water management condition, but not the variety, modulated the total antioxidant capacity (FRAP). This capacity was lowest at S2, while its values at S1 and S3 were similar to each other. The exception to this pattern was rainfed cv. Manzanilla, which presented increases with ripening. In both varieties, the greatest antioxidant capacity was found for the rainfed case. These results are at least partially similar to those reported by Arslan and Özcan [70], who described changes depending on ripening, environmental conditions and year. Petridis et al. [64], in the Gaidourelia, Kalamon, Koroneiki and Megalitiki varieties, described an increase in the antioxidant capacity under rainfed conditions, depending on the variety. The behaviour observed for the total antioxidant capacity (Figure 4) was very similar to that previously described for phenolic compounds, with full coincidence in S3, and dependence on the irrigation factor for both varieties. At each ripening stage, the antioxidant capacity was greater for the NI condition, reflecting a clear dependence of that parameter on water supplementation. These coincidences reflect the strong dependence of FRAP on the total content of phenolic compounds [64]. Nonetheless, the evolution of phenolic compounds and FRAP (Figure 4) was complementary to that presented by NADH oxidation, O_2_^−^ production as well as SOD and POX activities (Figure 3). The POX activity presented its greatest values at S2, as did SOD (but SOD maintained those values at S3), while the greatest values of the phenolic compounds were observed at S1 and S3.

## 4. Conclusions

The parameters studied depended on the stage of ripening, but for the same stage of ripening they were influenced by external factors to varying degrees. This was the case at S3, where the greater content of phenolic compounds and the greater antioxidant capacity strongly depended on irrigation (FI) or its absence (NI). Under rainfed conditions (NI), the content of these compounds was greater, and this factor was more determinant than the variety in terms of different trends in the parameters. At stages S1 and S2, there was no such clearly defined behaviour. In general, at S1 the variety factor was more determinant than the irrigation factor for phenolics. Similarly, the oxidant activities and the POX activity also depended on the additional supply of water, while the SOD and PPO activities through ripening were dependent on the variety, but not on the irrigation conditions. This analysis identified the variety as being the determining factor for the SOD and PPO activities, which are strongly involved in ripening. On the contrary, water supplementation was the factor determining the phenolic content, the antioxidant capacity and the oxidative activities. In addition, total phenolic content and antioxidant capacity were inversely correlated with the minimum temperature, while DW/FW ratio was inversely correlated with irrigation.

## Figures and Tables

**Figure 1 antioxidants-11-00729-f001:**
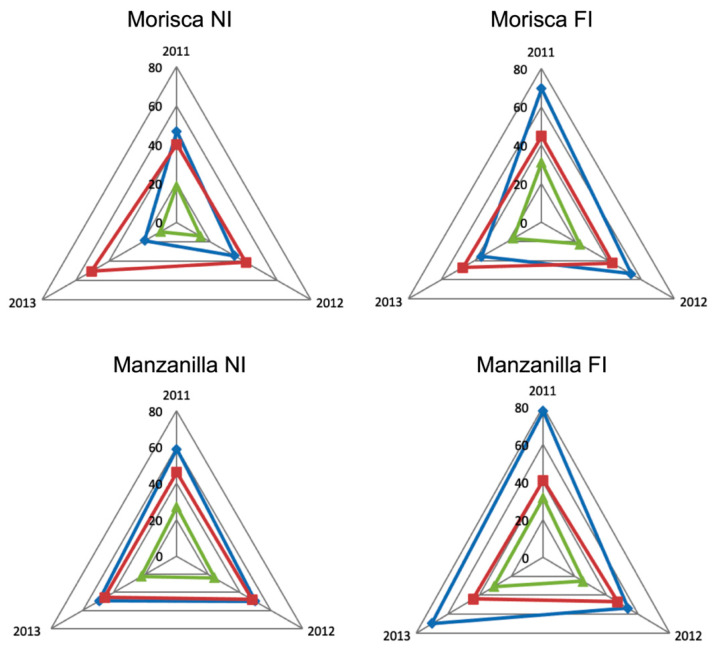
The olive fruit yield (kg ha^−1^ × 100) –♦–, oil concentration (% DW) –■– and yield (kg ha^−1^ × 100) –▲– in the varieties Morisca and Manzanilla, for rainfed (NI) and irrigated (FI) water regimes, during the three seasons studied (2011, 2012, 2013)—derived from Appendix A.

**Figure 2 antioxidants-11-00729-f002:**
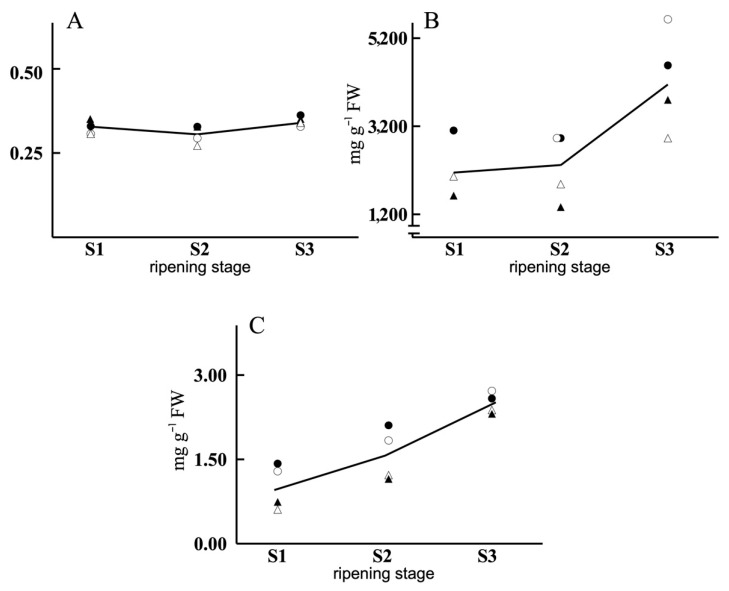
Line plots corresponding to DW/FW (**A**), total amino acids (**B**) and proteins (**C**) of olives of Morisca NI –●–, Morisca FI –○–, Manzanilla NI –▲– and Manzanilla FI –△–, for each ripening stage (derived from Appendix A).

**Figure 3 antioxidants-11-00729-f003:**
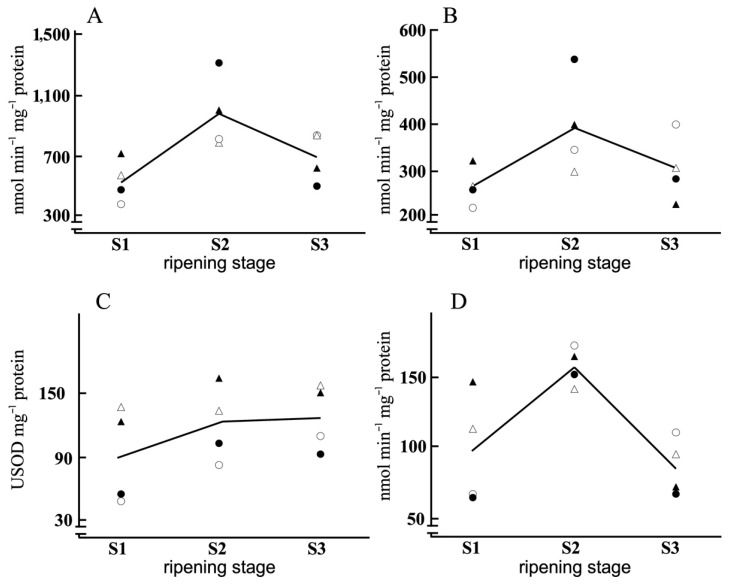
Line plots corresponding to NADH oxidation (**A**), O_2_^−^ production (**B**) and SOD (**C**) and POX (**D**) activities of olives of Morisca NI –●–, Morisca FI –○–, Manzanilla NI –▲– and Manzanilla FI –△–, for each ripening stage (derived from Appendix A).

**Figure 4 antioxidants-11-00729-f004:**
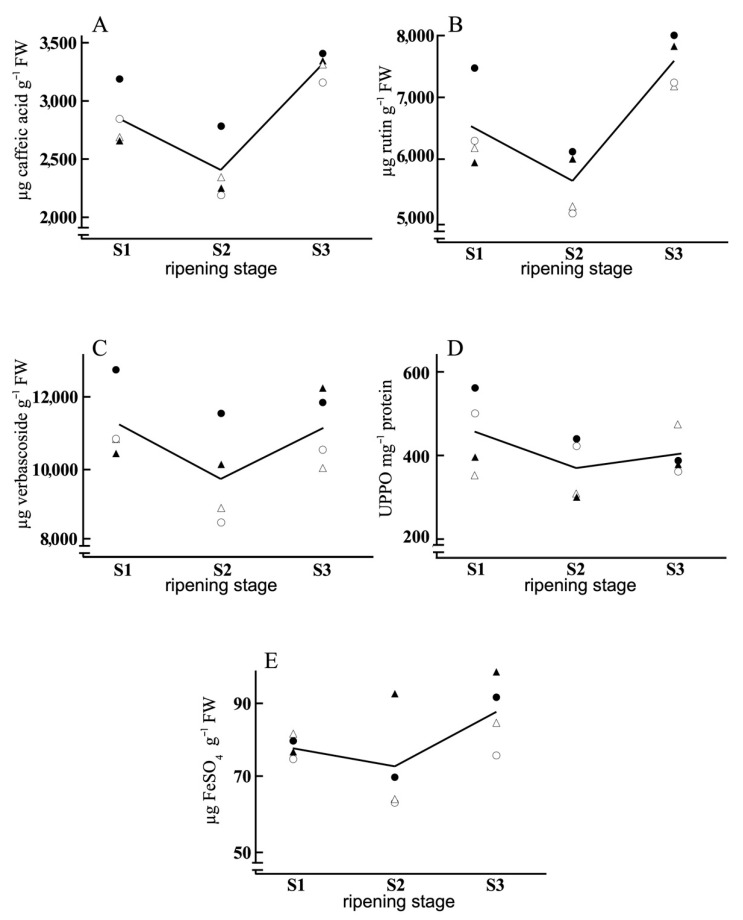
Line plots corresponding to total phenol (**A**), flavonoid (**B**) and PPG (**C**) contents; PPO activity (**D**); and FRAP (**E**) of olives of Morisca NI –●–, Morisca FI –○–, Manzanilla NI –▲– and Manzanilla FI –△–, for each ripening stage (derived from Appendix A).

**Table 1 antioxidants-11-00729-t001:** Mean values of the content parameters and estimates (*p*-values) of the influencing factors (derived from Appendix A).

Parameter	Mean	Cultivar	Irrigation	Ripening
DW/FW	0.30 ± 0.04		0.004	
Total amino acids (mg g^−1^ FW)	2866.35 ± 1637.10	8.28 × 10^−5^		1.46 × 10^−5^
Total protein (mg g^−1^ FW)	1.73 ± 0.83	4.64 × 10^−6^		1.64 × 10^−9^
NADH oxidation (nmol min^−1^ mg^−1^ protein)	730.74 ± 395.62			0.039
O_2_^−^ production (nmol min^−1^ mg^−1^ protein)	324.46 ± 174.50			
SOD activity (USOD mg^−1^ protein)	112.93 ± 54.87	3.06 × 10^−5^		
POX activity (nmol min^−1^ mg^−1^ protein)	113.27 ± 57.17			0.002
Total phenols (µg caffeic acid g^−1^ FW)	2856.15 ± 847.18			3.06 × 10^−6^
Total flavonoids (µg rutin g^−1^ FW)	6609.48 ± 1571.89			0.001
Total PPGs (µg verbascoside g^−1^ FW)	10,706.68 ± 2572.43		0.007	0.032
PPO activity (UPPO mg^−1^ protein)	403.37 ± 137.06	0.016		
FRAP (µg FeSO_4_ g^−1^ FW)	79.53 ± 42.16		0.031	0.002

**Table 2 antioxidants-11-00729-t002:** Average of the three years and seasonal variation index corresponding to each year (derived from Appendix A).

Parameter	Average	Seasonal Variation Index		
		2011	2012	2013
DW/FW	0.30 ± 0.02	0.01	0.00	−0.02
Total amino acids (mg g^−1^ FW)	2866.35 ± 1094.85	807.90	156.87	−964.78
Total protein (mg g^−1^ FW)	1.73 ± 0.77	0.13	0.02	−0.15
NADH oxidation (nmol min^−1^ mg^−1^ protein)	730.74 ± 222.74	−164.79	207.70	−42.90
O_2_^−^ production (nmol min^−1^ mg^−1^ protein)	324.46 ± 67.76	−97.23	142.93	−45.70
SOD activity (USOD mg^−1^ protein)	112.93 ± 20.04	−33.64	8.68	24.96
POX activity (nmol min^−1^ mg^−1^ protein)	113.27 ± 37.83	−7.12	28.50	−21.38
Total phenols (µg caffeic acid g^−1^ FW)	2856.15 ± 452.13	−901.98	483.20	418.78
Total flavonoids (µg rutin g^−1^ FW)	6609.48 ± 973.57	−1132.08	965.16	166.92
Total PPGs (µg verbascoside g^−1^ FW)	10,706.68 ± 884.33	−2603.04	1435.53	1167.51
PPO activity (UPPO mg^−1^ protein)	403.37 ± 43.72	118.41	−44.77	−73.64
FRAP (µg FeSO_4_ g^−1^ FW)	79.53 ± 7.56	−54.16	37.31	16.86

**Table 3 antioxidants-11-00729-t003:** Significant correlations and *p*-values (in parentheses) between parameters and environmental factors.

	Flavonoids	PPGs	FRAP	Total Amino Acids	POX	NADH Oxidation	SOD	PPO	DW/FW	T_min_	T_max_	Irrigation
Total phenols	0.894 (0.001)	0.889 (0.001)	0.866 (0.003)							−0.687 (0.041)	−0.869 (0.002)	
Total flavonoids		0.863 (0.003)	0.729 (0.026)								−0.823 (0.006)	
Total PPGs			0.872 (0.002)					−0.689 (0.040)			−0.801 (0.009)	
FRAP											−0.829 (0.006)	
Total protein				0.838 (0.005)								
Total amino acids									0.807 (0.009)			
O_2_^−^ production					0.793 (0.011)	0.882 (0.002)						
POX						0.876 (0.002)						
NADH oxidation							0.691 (0.039)					
DW/FW												−0.483 (0.007)

## Data Availability

The original contributions generated for this study are included in the article/Appendix A; further inquiries can be directed to the corresponding author.

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
