# Peer review of "Effect of Water Supplementation on Oxidant/Antioxidant Activities and Total Phenol Content in Growing Olives of the Morisca and Manzanilla Varieties"

_antioxidants, 2022, doi:10.3390/antiox11040729_

Round 1
Reviewer 1 Report
The manuscript (antioxidants-1621287) “Effect of water supplementation on oxidant/antioxidant activities and total phenols content in growing olives of the Morisca 3 and Manzanilla varieties” analyzed the effect of the water regime (with or without irrigation) in two olive cultivars (Manzanilla and Morisca), at three ripening stages (S1, S2, S3) and during three consecutive years. Overall, the design and implementation of the experiment are very good. However, when I thought about the novelty of this research, I found that there were so many similarities in a research report published by the same author (Espinosa F, 2019), which is listed below. There are a large number of identical data tables. And the author did not quote this article in the manuscript.
Sáinz JA, Garrido I, Hernández M, Montaño A, Llerena JL, Espinosa F (2019) Influence of cultivar, irrigation, ripening stage, and annual variability on the oxidant/antioxidant systems of olives as determined by MDS-PTA. PLoS ONE 14(4): e0215540.
https://doi.org/10.1371/journal.pone.0215540
Author Response
We would like to thank for improving so much our MS and also appreciate your suggestions. We hope that the MS can be now considered suitable for publication in Special Issue.
This ms is a continuation of the previous one, which was a first approximation to the evolution of a series of parameters and their interaction, based on an MDS-PTA analysis. The present ms is a new approach, in which, starting from the primary data obtained by our group, a new methodology is used based on time series analysis. Indeed the climatic data shown in table S1 and the primary data used in this article come from the sampling campaigns carried out in 2011-2012 and 2013, and the biochemical determinations carried out, tables S2 and S3. Logically in the table headings the reference of that paper is missing. Apologies.
In this paper the objective is not to explain the behavior of the parameters in the different experimental conditions. Now we intend to use time series analysis as a study tool and a means of discrimination in works with a large amount of data and many factors that influence and interrelate with each other. This analysis allows a clearer and simpler approach to the identification of the determining factor in each of the parameters studied, eliminating interferences.
MDS-PTA is a multivariate analysis that helps to visualize changes in each variable, based on principal component analysis. It allows searching for new variables that simplify the complexity of the relationships between variables and individuals for analysis. However, as time series analysis is a univariate technique, the pattern of change of each real variable with respect to individuals is obtained. This allows us to perform a different, and simpler, type of analysis based on the original variables.
Thank you for helping us to improve our MS.
Sincerely yours,
The authors
Reviewer 2 Report
The manuscript “Effect of water supplementation on oxidant/antioxidant activi-2 ties and total phenols content in growing olives of the Morisca 3 and Manzanilla varieties” studies the effect of the water regime (with or without irrigation) in two olive cultivars, at three ripening stages and during three consecutive years.
There is no doubt that this is a laborious and time-consuming study. The article is well organised, and the methods developed are appropriate. In addition, a good statistical analysis of the results obtained in the three years is carried out, which allows the data to be modelled to draw conclusions for future years on the same olive crops.
Without any doubt it is an interesting topic to be studied, However, some aspects should be considered before publication. Find below my recommendations:
Abstract: I recommend improve your abstract by adding the effective correlation among factors you describe. I mean, by including percentages for the improve of analysed parameters of R values for correlations.
Keywords: I recommend that you do not include words that already appear in the title.
Material & Methods:
Section 2.3. In addition to citing the methods that have been carried out, could you describe them (adding reagents and the equipment used for their determination)? For instance, why did you use caffeic acid as standard to measure total phenolic content? Is that the main phenolic found in olives? Please, clarify and add more information for the lector.
Section 2.4. How did you obtain the extract of the raw olives?
Section 2.5. There are several methods to measure the antioxidant activity, why did you choose the FRAP method? Some of the functional groups of olive pulp and peels can form complexes with iron and cause precipitates that may alter the absorbances obtained by this method. Did you compare these measurements with other methods?
Please, add to all the sections of the M&M the spectrophotometer (brand, city, country) used to perform your methods. It was always the same equipment, add it the first time that you cite it.
Results & Discussion:
Although the authors approach the subject in such a way that they look for correlations between the values obtained, before the statistical approach, from my point of view, it seems that the description of the results is missing, detailing the level of improvement or change that the irrigation entails, specifying it in percentages or detailing the increase that is produced.
Even though the values obtained are old (dating back 10 years), the article is interesting given the statistical approach and the modelling of the results. Is anything known about today's crops? Could something be added at the end of the results/conclusions? Can these values be taken into account for the future?
Supplementary material: These tables result interesting for the reader, but they are very difficult to read.
Table S1. It is difficult to read. Please, fit the table on a horizontal sheet and adds more space between the values and increase the font of the letter.
Table S3. The same than Table S1. It is impossible to read in font size 6.
Regarding Table S2 and Table S3: Did you statically analyse these data? There were significant differences between irrigation treatments? Or varieties? Or collection year? I miss these data in the body of the manuscript.
Do you have any picture of the crop/collection/analysis? A graphical abstract or a picture on the graphics would make the paper more attractive for the reader.
Author Response
We would like to thank for improving so much our MS and also appreciate your suggestions. We hope that the MS can be now considered suitable for publication in Special Issue.
Abstract
Thank you for your suggestion, has been added correlations values.
Keywords
Thank you for your suggestion, we have removed the keywords included in the title.
Material y methods
Section 2.3
We have used caffeic acid to perform the standard curve, in the way described in the method of determination of total phenols used. It is possible to use other compounds of this type, e.g. gallic acid (Hajimahmoodi et al., 2008; Petridis et al., 2012; Stamapoulos et al., 2013; Soufi et al., 2016; Gouvinhas et al., 2017; Debbou-louknane et al., 2019; Gómez-Cruz et al., 2020; Marinopoulou et al., 2020;). This compound is used to perform the standard curve in the same reaction in which all phenolic compounds produced in the presence of the Folin-Ciocalteau reagent are measured. In this reaction the amount of phenolic compounds present in an extract is measured, since in alkaline pH conditions and in the presence of the Folin-Ciocalteau reagent they react with it as oxidizing agents, producing a coloration that can be determined spectrophotometrically at 765 nm. This and the other methods are described briefly, so as not to make this section too long. Caffeic acid also forms part of the phenolic profile of olives. More details are given in the corresponding ms references.
Section 2.4
The pulp of the olives sampled is used in all cases. This has been detailed more clearly in the text.
Section 2.5
In our material we have not observed any precipitation or turbidity in the reaction medium. This method is widely used in this type of material (Hajimahmoodi et al., 2008; Petridis et al., 2012; Gouvinhas et al., 2017; Gómez-Cruz et al., 2020; Marinopoulou et al., 2020).
We have included the brand, model, city, and country of the spectrophotometer used.
Results and discussion
In the discussion we have focused primarily on the behavior of the data in relation to the different variables, explaining the contribution that this type of time series analysis allows us to obtain. A more detailed description would have made the article very long. The objective is to identify the behavior of the different parameters determined as a function of the different variables, and to establish which of them controls each parameter. This can allow us to know in advance the characteristics that the final product of the crop will have according to the factors that influence its production at a specific moment (variety, ripening stage, climatic conditions, irrigation contribution, etc.).
The primary data were obtained from material collected in the 2011, 2012 and 2013 seasons. The two olive grove plots are commercially exploited, and continue to be cultivated, but unfortunately we no longer have data from them. The owner company completed the project in which the study was framed.
Nevertheless we consider that, according to the data obtained, the behavior will have been very similar to the one described, with the fluctuations derived from the state of maturation and climatology of each campaign. Recent years have been very similar in terms of temperature and rainfall. The main contribution of our study is that it is a tool that allows us to know how these varieties will respond to irrigation or water deficit in terms of phenolic composition, and its dependence on the ripening stage. This can be used to harvest each variety at a different ripening time in response to external factors, and obtain different virgin olive oils with different qualities.
Supplementary material
We agree with your assessment, the tables are large and difficult to read and interpret directly. This is precisely what justifies the statistical approach used in the article.
If this article is accepted, we will request in the editing process that the tables be placed in a vertical position, which will allow an increase in the font size.
Tables S2 and S3 have been used as the basis for our time series analysis.
The individual analysis is already included in another paper (Sainz et al., 2019). The objective of this paper is to see the behavior of the parameters analyzed for each combination of cases (variety, irrigation, maturity) and to compare them in case there are coincidences.
Thank you for the thorough review of our article, and for your invaluable contribution in clarifying and improving it.
Thank you for helping us to improve our MS.
Sincerely yours,
The authors

Reviewer 3 Report
Line 28-29 …remove from abstract
Line 84 ..Which of the time series analyses methods was applied?
Reviewer comment:“ Time series analysis includes methods (several methods!!!) for analyzing time series data in order to extract significant statistics factor and other data characteristics. „
Line 169-172 ..Instead: Three types of analysis were carried out, two exploratory and one analytical. In the first, spider graphs were applied to compare each variety's productive parameter profiles according to the water management regime. In the second, a correlation analysis was performed between the chemical compounds and the climatological data.
It should be written: “Spider graphs were applied to compare each variety's productive parameter profiles according to the water management regime. The correlations between the chemical compounds and climatological data were calculated” A slightly modified….. ……etc.
Line 191.. The discussion should be separated from the results.
Line 225.. instead of p values it is common to show correlation coefficients (Pearson…Spearman) and p values.
Line 230: Table 1 …instead of p values it is common to show correlation coefficients and p values. The correlation coefficients are commented together with the corresponding p values
Line 248..instead : The multiple interactions between phenolic compounds, antioxidant activ- 248 ities, antioxidant capacity, rainfall, temperatures, insolation, soils, etc. were subjected to a 249
correlation analysis, and the results are presented in Table 3.
It should be written: The multiple interactions between phenolic compounds, antioxidant activ- 248 ities, antioxidant capacity, rainfall, temperatures, insolation, soils, etc. were presented by correlation coefficient and the results are presented in Table 3.
Line 257… significance level p <0.05 or? p<0.01 ?
- Line 503-513 …Remove from text
Author Response
We would like to thank for improving so much our MS and also appreciate your suggestions. We hope that the MS can be now considered suitable for publication in Special Issue.
We have modified the abstract (eliminated lines 28-29) and also included your suggestions (lines 169-172; 248 and 503-513).
Line 84. Within linear processes, a simple additive model was applied as method of time series analysis.
Lines 225 and 230. Pearson correlation analysis was carried out by SPSS program, and p-value associated with that correlation analysis was used. The p-value was adde to the table 3.
We have combined results and discussion because we think it is easier to understand. The tables and figures are very necessary in the discussion, and in this manner they are more accessible.
Thank you for helping us to improve our MS.
Sincerely yours,
The authors

Round 2
Reviewer 1 Report
The author mentioned in the revised manuscript that different analysis methods were used for the previous data. In this case, the author should quote the publish article in the introduction at the beginning to point out the novelty of the research. With the same data and different analysis methods, I personally think this manuscript can be published in other relevant journals, which is not very suitable for Antioxidants.
Author Response
We have included the citation of the previous ms with whose data we are working, at the beginning of material and methods.
To analyze the data of a trial, the mean values and the result of ANOVAs are used to find out if there have been differences between them. This is easy when trying to analyze the effect of a single factor. When there are more factors, the observations become increasingly complicated. However, it is still common to use the result of mean values obtained from the combination of factors, and the result of MANOVAs for the same purpose. Multivariate analysis is often used to facilitate the analysis.
In this paper, we have proposed another method of comparing results based on the shape of the graphs. These graphs are constructed with mean values and collect all the factors, thus facilitating the understanding of the results.
The objective is not to show a new statistical tool. The time series analysis used is not new. The novelty is to use the graphs to compare the behavior of the different compounds and to explain the observed changes.
This ms can serve as a reference for other similar studies in other crops in which the evolution of parameters with development, ripening, their dependence and interrelation, environmental and agronomic factors, etc., are studied, so we consider it suitable for publication in Antioxidants.